# Is Maternal Selenium Status Associated with Pregnancy Outcomes in Physiological and Complicated Pregnancy?

**DOI:** 10.3390/nu16172873

**Published:** 2024-08-27

**Authors:** Joanna Pieczyńska, Sylwia Płaczkowska, Rafał Sozański, Halina Grajeta

**Affiliations:** 1Department of Dietetics and Bromatology, Wroclaw Medical University, Borowska 211, 50-556 Wroclaw, Poland; halina.grajeta@umw.edu.pl; 2Diagnostics Laboratory for Teaching and Research, Wroclaw Medical University, Borowska 211a, 50-556 Wroclaw, Poland; splacz@op.pl; 3Department and Clinic of Gynecology and Obstetrics, Wroclaw Medical University, Borowska 213, 50-556 Wroclaw, Poland; rafal.sozanski@umw.edu.pl

**Keywords:** selenium, GPX, pregnancy, pregnancy complications, pregnancy outcomes

## Abstract

Selenium is essential for the synthesis and function of various selenoenzymes, such as glutathione peroxidases, selenoprotein P, and thioredoxin reductase. These enzymes play a critical role in both antioxidant defense and in limiting oxidative damage. Numerous studies have reported associations between serum selenium concentration, obstetric complications and pregnancy outcomes. The aim of this study was to determine whether the dietary intake of selenium, its serum concentration, and the activity of glutathione peroxidase in subsequent trimesters of pregnancy affect the birth condition of newborns. This was assessed based on the APGAR score in the 1st and 5th minute of life, birth weight, body length and head and chest circumference in both physiological and complicated pregnancy courses. Twenty-seven pregnant women, with a mean age of 29.6 ± 4.8 years from the Lower Silesia region of Poland, participated in the study. Fifty-five percent of the study group experienced pregnancy complications. The median reported selenium intake and serum selenium content for Polish pregnant women in the first trimester was 56.30 μg/day and 43.89 μg/L, respectively. These figures changed in the second trimester to 58.31 μg/day and 41.97 μg/L and in the third trimester to 55.60 μg/day and 41.90 μg/L. In the subgroup of pregnant women with a physiological pregnancy course, a weak, positive correlation was observed in the first trimester between Se intake and the length (R = 0.48, *p* = 0.019) and the birth weight of newborns (R = 0.472, *p* = 0.022). In the second trimester, a positive correlation was noted with the APGAR score at the 1st (R = 0.680, *p* = 0.005) and 5th minutes (R = 0.55, *p* = 0.033), and in the third trimester with the APGAR score at the 1st minute (R = 0.658, *p* = 0.019). The glutathione peroxidase activity had a strong positive correlation with the APGAR score at the 1st min (R = 0.650, *p* = 0.008) in the second trimester and with the birth weight of the newborns (R = 0.598, *p* = 0.039) in the third trimester. No correlation was found between newborns’ birth measurements and serum selenium concentration. In the subgroup of pregnant women with complications, a strong, negative correlation was found between Se intake in the second trimester and gestational age (R = −0.618, *p* = 0.032). In the third trimester, a positive correlation was noted between Se concentration in serum and head circumference (R = 0.587, *p* = 0.021). The results indicate that maternal selenium status during pregnancy, including dietary intake, serum concentration, and glutathione peroxidase activity, correlates with anthropometric parameters of the newborn, such as birth weight, length, and APGAR score, especially in pregnancies with a physiological course. However, these relationships diminish in importance when pregnancy complications occur.

## 1. Introduction

An adequate supply of macro- and microelements is crucial for the proper functioning of the human body, especially for pregnant women. A well-balanced diet significantly impacts the course of pregnancy, fetal development, and the health of both the child and the pregnant woman. Through food, the mother’s body absorbs all nutrients, including trace elements, which are transported across the blood–placenta barrier to the fetus. Selenium is essential for the synthesis and function of various selenoenzymes such as glutathione peroxidases, selenoprotein P, and thioredoxin reductase. These enzymes play vital roles in antioxidant defense and in reducing oxidative damage. Selenium also inhibits the activation of carcinogens, prevents the accumulation of DNA damage and lipid peroxidation, and suppresses the induction of pro-inflammatory cytokines [1,2,3]. Previous studies have shown that plasma Se concentrations and glutathione peroxidase activity decrease during pregnancy, partly due to the increasing erythrocyte mass in the developing fetus [4,5].

Optimal selenium levels may reduce the risk of complications for both the pregnant woman and the fetus, such as miscarriages, thyroid dysfunction, preeclampsia, premature birth, and gestational diabetes [6]. Selenium deficiency in pregnant women can lead to nervous system dysfunction in the developing fetus. Oxidative stress significantly affects the gestational parameters and can cause intrauterine growth retardation and abnormal tissue development, linked to endocrine metabolic imbalances. Research indicates that low selenium intake is also associated with the risk of giving birth to infants with low birth weight or those small for gestational age. In Poland, the soil’s low selenium content is reflected in its food concentration. Research by Filipowicz et al. [7] demonstrates that selenium intake in the diet of Polish pregnant women is lower than in other European countries.

To date, no studies have explored the impact of the selenium status—measured by intake, serum concentration and glutathione peroxidase activity—throughout pregnancy on the condition of the newborn. The proposed research aims to verify findings related to this topic.

The goal of the research is to determine whether dietary selenium intake, its serum concentration, and the activity of glutathione peroxidase in different trimesters of pregnancy influence the birth condition of newborns. This will be assessed using the APGAR score at the 1st and 5th minute of life, birth weight, body length, and head and chest circumference in both physiological and complicated pregnancy courses.

## 2. Material and Methods

### 2.1. Study Design

The study was designed as a pilot prospective cohort investigation involving Polish pregnant women aged 18–40 regardless of the BMI and week of pregnancy. The minimum sample size was determined based on an α error probability of 0.5, a power of 0.8, and a minimum effect size of 0.5 for a correlations test, yielding a required sample size of 21 patients. Eventually, 27 pregnant women could be recruited for the study based on the inclusion and exclusion criteria (see Figure 1). Post hoc power analysis for the correlation test, assuming a minimum effect size of r = 0.5, an α error probability of 0.05, and a sample size equal to 12 for the physiologic pregnancy group, yielded a power of 0.68. For the complicated pregnancy group of 15 with the same assumptions, the power of analysis was 0.59. These calculations were performed using the online G*Power calculator (v. 3.1.9.6, developed by Franz Faul, Universitas Kiel, Germany, https://www.psychologie.hhu.de/arbeitsgruppen/allgemeine-psychologie-und-arbeitspsychologie/gpower, accessed on 24 June 2024).

### 2.2. Participants

This prospective study was conducted from November 2013 to February 2017 at the Department of Food Science and Dietetics of Wroclaw Medical University. A total of 94 pregnant women were recruited from several private maternity clinics across Lower Silesia. Eligibility for the study excluded pre-existing conditions such as hypothyroidism, hypertension, diabetes, autoimmune diseases, cardiovascular diseases, and recurrent cystitis. Women with multiple pregnancies were also excluded from the study. Having more than one fetus creates a higher metabolic demand than a single pregnancy, which may increase the risk of nutrient deficiencies. All participants provided informed, written consent based on ethical approval from the Bioethics Committee of the Medical University of Wrocław KB-884/2012 (20 December 2012). Obstetric care and supervision of the study participants were provided by private maternity clinics until delivery, where any pregnancy complications were documented in the medical records. In the studied group of pregnant women, serious pregnancy complications were recorded, such as gestational hypothyroidism (presence of elevated serum TSH with normal fT4 or TT4 values—subclinical hypothyroidism or decreased fT4 or TT4 values—over hypothyroidism referring to valid reference ranges for TSH and thyroid hormones established by medical centers for a local population of healthy, iodine-sufficient, thyroid peroxidase antibody (TPOAb)-negative pregnant women without thyroid illness) and gestational diabetes (glucose levels over 140 mg/dL in the oral glucose tolerance test), as well as minor complications such as anemia (according to World Health Organization (WHO) hemoglobin concentration < 110 g/L) and urinary tract infections. No other pregnancy complications were reported. The study design assumed recruitment of participants regardless of the week of pregnancy, but the vast majority of participants joined the study in the second trimester of pregnancy. Of a total of ninety-four pregnant women recruited, only thirty-two subjects participated throughout all three trimesters of pregnancy, of which only twenty-seven provided documentation of the anthropometric parameters of their offspring (Figure 1). All pregnant women declared taking mineral and vitamin supplements containing or not containing selenium throughout their pregnancy. The characteristics of the study group are presented in Table 1 and Table 2.

### 2.3. Blood Samples

Blood was collected from study participants once in each trimester, i.e., 8–14 weeks of pregnancy (T I), 18–24 weeks of pregnancy (T II) and 31–36 weeks of pregnancy (T III). Under the supervision of medical staff, in accordance with established protocols, fasting blood was collected in the morning by venipuncture into S-Monovette^®^ tubes.

### 2.4. Biochemical Measurements

#### Glutathione Peroxidase (GPX)

The assessment of GPX activity in whole blood was performed using the RANSEL Randox^®^ kit (Randox Laboratories, Ltd., Crumlin, UK) on the Konelab 20i autoanalyzer (ThermoScientfic, Vantaa, Finland). GPX catalyzes the oxidation of glutathione by cumene hydroperoxide. In the presence of glutathione reductase and NADPH, the oxidized glutathione is immediately converted to its reduced form with a concomitant oxidation of NADPH to NADP+. The decrease in absorbance at 340 nm was expressed as units per liter of the hemolysate.

### 2.5. Selenium Intake

The assessment of selenium intake was based on measurements using electrothermal atomic spectrometry with Zeeman’s background correction (AAS Z-5000, Hitachi, Tokyo, Japan) in reconstituted food rations. Once in each trimester of pregnancy, study participants provided a seven-day food diary, along with samples of the food they consumed and dietary supplements they took, a week before blood collection. To avoid memory errors, each record was verified by a trained interviewer in terms of composition and quantity of recorded products and cross-referenced with the food samples provided. On the basis of the seven-day food diary and collected food samples, the average 24 h food ration was reconstructed and homogenized. The Se concentration was determined in duplicate from homogenates of reconstituted daily food rations weighing 4 g. Samples of the homogenate were weighed into Teflon reaction vessels to which 5 mL of 70% HNO_3_ (Baker, Instra-Analyzed for trace elements, Sanford, ME, USA) was added. They were digested in a Milestone mls 1200 mega microwave device (Milestone™ Srl, Sorisole, Italy). Parallel blanks containing the amounts of reagents indicated above were prepared. To assess the accuracy of the method, a certified reference material was used—Simulated Diet D (LivsmedelsVerked National Food Administration, Uppsala, Sweden). The estimated total selenium intake (as an element in inorganic form) for each study participant was presented as an average over seven days.

The daily intake of selenium takes into account the amount of this element supplied with supplements. The supplements contained selenium in amounts ranging from 20 to 60 µg/day.

The Se intake adjusted for energy supply was also calculated. The mean daily intake of energy was computed from 24 h recalls from 7 consecutive days in each trimester using a tailored computer program Dieta 6.0.

### 2.6. Serum Selenium Determination

Selenium concentration was determined directly in serum by electrothermal atomic spectrometry with Zeeman background correction (AAS Z-5000 Hitachi, Tokyo, Japan). The accuracy of the method was assessed on the reference material: Seronorm Trace Element Serum L-1 (Nycomed AS, Oslo, Norway).

### 2.7. Pregnancy Outcome Data

Measures describing pregnancy course, obstetric complications, and neonatal outcome status were obtained from either the medical records or through subject self-report. Based on the medical records provided by mothers after delivery the following data were obtained regarding mode of delivery, gestational age at birth, APGAR score at 1 and 5 min after birth, birth weight, body length, head circumference and chest circumference.

### 2.8. Statistical Analysis

Due to the small number of participants in groups for individual maternal complications in subsequent trimesters of pregnancy, it was not possible to perform reliable statistical analyses; therefore, all complications were included as one group. Similarly, too low a number of pregnant women with complications in the first trimester did not allow for a correlation analysis to be performed. The normality of variable distributions was assessed using three different statistical tests: the Kolmogorov–Smirnov test, the Lillefors test and the Shapiro–Wilk test. Nonparametric analysis of variance with the Friedman post hoc test for dependent variables was used to compare between trimesters GPX activity, serum selenium concentration, and Se daily intake and per 1000 kcal. Mann–Whitney U tests were used to compare concentrations/neonatal outcomes in the physiological and complicated pregnancy groups. The Spearman rank correlation test was used to assess correlations between selenium status and anthropometric parameters of newborns. For all tests, the confidence level of α = 0.05 was considered statistically significant. The data were analyzed using Statistica 13.1, PL (Statsoft). The χ^2^ Fisher exact test analysis for this paper was generated using the Real Statistics Resource Pack software (Release 8.9.1 (Copyright (2013–2023) Charles Zaiontz; www.real-statistics.com, accessed on 20 August 2024).

## 3. Results

### 3.1. Participant Baseline Characteristics (Table 1 and Table 2)

Mean maternal age was 29.6 ± 4.8 years, and most of the study participants (74.1%) were in the range of 26–34 years. With each subsequent trimester of pregnancy, the rate of pregnancy complications increased from 14.8% in the first trimester to 55.6% in the third trimester (Table 1). There was only one case of premature birth (<37 weeks of pregnancy) and low birth weight (<2500 g) among study participants (Table 2). Almost all surveyed pregnant women declared education on academic level, lived in an urban area and never smoked cigarettes. All study participants used vitamins, minerals or mixed supplements dedicated for pregnant women, but only 29.6% of the group took supplements containing selenium. The mean pre-pregnancy BMI value was 22.09; however, three study participants were malnourished and five were overweight, and no pregnant women were obese before pregnancy. In most cases, the physical activity of the participants was at the sedentary level throughout the entire pregnancy (Table 1). When comparing the characteristics of study participants with a physiological course of pregnancy to that with a complicated course, no significant differences were observed between the groups (Table 2). Only in the case of pre-pregnancy BMI, in the group with pregnancy complications, more cases of BMI outside the norm were recorded, and more respondents took vitamin and mineral supplements with Se.

### 3.2. Selenium Status during Gestation (Table 3)

In the second trimester of pregnancy, the highest selenium intake, as well as the energy-adjusted Se intake and the highest GPX value, as opposed to the serum Se concentration, were observed; however, significant differences were not observed between values for all parameters at particular trimesters.

**Table 3 nutrients-16-02873-t003:** Selenium status during pregnancy, median (Q1–Q3).

	T I *n* = 27	T II *n* = 27	T III *n* = 27	*p*-Value
Se intake [µg/day]	56.30 (37.42–58.97)	58.31 (43.24–73.95)	55.60 (41.54–62.99)	NS
Se intake [µg/1000 kcal]	26.05 (21.00–31.71)	30.49 (20.71–34.43)	24.71 (20.76–31.10)	NS
Se serum [µg/L]	43.98 (39.05–48.05)	41.97 (33.22–49.56)	41.90 (37.7–48.14)	NS
GPX [U/L]	217 (200–242)	233 (207–257)	231 (195–255)	NS

NS—not significant; T I—the first trimester of pregnancy; T II—the second trimester of pregnancy; T III—the third trimester of pregnancy; Se—selenium; GPX—glutathione peroxidase.

### 3.3. Mean Values of Neonatal Outcome Measurements and Selenium Status in Normal and Complicated Pregnancy (Table 4)

Pregnant women with pregnancy complications were characterized only by significantly lower serum selenium concentration and higher gestational age of the newborns compared to healthy pregnant women.

**Table 4 nutrients-16-02873-t004:** Neonatal outcome measurements and selenium status in physiological and complicated pregnancy, median (Q1–Q3).

	Physiological Pregnancy *n* = 12	Pregnancy Complications *n* = 15	*p*-Value
Se intake [µg/day]	55.57 (41.15–59.86)	55.60 (41.54–62.99)	NS
Se intake [µg/1000 kcal]	25.26 (22.02–31.42)	25.98 (20.71–35.11)	NS
Se serum [µg/L]	43.97 (40.76–49.80)	40.19 (34.84–46.54)	0.029
GPX [U/L]	226 (199–233)	234 (209–260)	NS
Body length [cm]	53 (52–55)	54 (52–54)	NS
Chest circumference [cm]	33 (32–34)	34 (33–34)	NS
Head circumference [cm]	33 (31.5–34)	34 (33–34)	NS
Birth weight [g]	3355 (2815–3765)	3500 (3350–3600)	NS
APGAR score at 1 min	10 (9.5–10)	10 (9–10)	NS
APGAR score at 5 min	10 (10–10)	10 (10–10)	NS
Gestational age at birth [weeks]	39 (37.5–39.5)	40 (40–41)	0.005

NS—not significant; Se—selenium; GPX—glutathione peroxidase.

In all studied neonatal outcome measurements: birth weight, body length, head circumference and chest circumference, results were insignificantly higher in women with complicated pregnancies than with normal pregnancies.

### 3.4. Correlations Observed between Selenium Status Parameters and Neonatal Outcome Measurements during Normal and Complicated Pregnancy (Table 5)

In the subgroup of pregnant women with a physiological pregnancy course, a weak, although statistically significant, positive correlation was found in the first trimester between Se intake and the length and birth weight of newborns, Se-energy adjusted intake and the length and chest circumference of newborns as well as gestational age. Se intake correlated in the second trimester with the APGAR score at 1 and 5 min, as well as in the third trimester with the APGAR score at 1 min. Furthermore, energy-adjusted Se intake correlated with the APGAR score at 1 min. The GPX value had a strong positive correlation with the APGAR score at 1 min in the second trimester and with the birth weight of the newborns in the third trimester. There was no correlation between newborns’ birth measurements and serum selenium concentration.

**Table 5 nutrients-16-02873-t005:** Correlations observed between selenium status parameters and neonatal outcome measurements during physiological and complicated pregnancy.

Correlations	T I		T II		T III	
	R	*p*-Value	R	*p*-Value	R	*p*-Value
Physiological pregnancy						
Se intake vs. body length	**0.481**	**0.019**	0.090	0.748	0.047	0.884
Se intake vs. birth weight	**0.472**	**0.022**	0.017	0.949	0.316	0.315
Se intake vs. chest circumference	0.379	0.074	−0.060	0.830	−0.071	0.825
Se intake vs. head circumference	0.247	0.255	−0.247	0.374	0.111	0.730
Se intake vs. APGAR score at 1 min.	0.090	0.680	**0.680**	**0.005**	0.658	0.019
Se intake vs. APGAR score at 5 min.	0.119	0.586	**0.551**	**0.033**	0.573	0.051
Se intake vs. gestational age	0.093	0.671	−0.153	0.584	−0.279	0.379
Se intake/1000 kcal vs. body length	**0.500**	**0.015**	0.014	0.963	−0.075	0.815
Se intake/1000 kcal vs. birth weight	0.382	0.071	0.043	0.886	0.272	0.391
Se intake/1000 kcal vs. chest circumference	**0.550**	**0.006**	0.014	0.963	−0.060	0.852
Se intake/1000 kcal vs. head circumference	0.372	0.079	−0.353	0.235	0.169	0.598
Se intake/1000 kcal vs. APGAR score at 1 min.	−0.321	0.135	0.301	0.316	**0.685**	**0.013**
Se intake/1000 kcal vs. APGAR score at 5 min.	−0.368	0.083	0.231	0.446	0.499	0.097
Se intake/1000 kcal vs. gestational age	**0.466**	**0.024**	−0.205	0.501	−0.212	0.507
Se serum vs. body length	0.046	0.832	−0.064	0.819	0.285	0.367
Se serum vs. birth weight	0.033	0.878	0.078	0.779	0.239	0.453
Se serum vs. chest circumference	−0.018	0.931	−0.027	0.922	0.203	0.524
Se serum vs. head circumference	0.216	0.321	−0.122	0.663	0.200	0.532
Se serum vs. APGAR score at 1 min.	0.150	0.494	0.212	0.447	0.496	0.100
Se serum vs. APGAR score at 5 min.	0.122	0.578	−0.133	0.635	0.286	0.365
Se serum vs. gestational age	−0.108	0.622	−0.035	0.900	−0.213	0.504
GPX vs. body length	0.318	0.138	0.243	0.381	0.231	0.469
GPX vs. birth weight	0.155	0.478	0.334	0.222	**0.598**	**0.039**
GPX vs. chest circumference	−0.008	0.970	−0.022	0.936	0.225	0.481
GPX vs. head circumference	0.212	0.329	0.173	0.537	0.155	0.628
GPX vs. APGAR score at 1 min.	0.132	0.545	**0.650**	**0.008**	0.055	0.863
GPX vs. APGAR score at 5 min.	−0.033	0.880	0.497	0.058	−0.108	0.737
GPX vs. gestational age	−0.182	0.404	−0.116	0.679	0.025	0.937
Complicated pregnancy						
Se intake vs. body length	x	x	−0.281	0.375	−0.133	0.634
Se intake vs. birth weight	x	x	−0.394	0.204	−0.322	0.240
Se intake vs. chest circumference	x	x	−0.400	0.197	−0.134	0.633
Se intake vs. head circumference	x	x	−0.102	0.752	−0.168	0.548
Se intake vs. APGAR score at 1 min.	x	x	−0.142	0.658	0.142	0.611
Se intake vs. APGAR score at 5 min.	x	x	−0.091	0.777	−0.012	0.965
Se intake vs. gestational age	x	x	**−0.618**	**0.032**	−0.083	0.766
Se intake/1000 kcal vs. body length	x	x	−0.337	0.259	−0.135	0.630
Se intake/1000 kcal vs. birth weight	x	x	−0.381	0.198	−0.458	0.085
Se intake/1000 kcal vs. chest circumference	x	x	−0.252	0.405	−0.231	0.407
Se intake/1000 kcal vs. head circumference	x	x	−0.121	0.691	−0.060	0.830
0Se intake/1000 kcal vs. APGAR score at 1 min.	x	x	−0.178	0.558	−0.124	0.658
Se intake/1000 kcal vs. APGAR score at 5 min.	x	x	−0.069	0.820	−0.060	0.830
Se intake/1000 kcal vs. gestational age	x	x	−0.531	0.061	−0.218	0.433
Se serum vs. body length	x	x	0.060	0.851	−0.045	0.871
Se serum vs. birth weight	x	x	0.309	0.327	0.039	0.889
Se serum vs. chest circumference	x	x	0.392	0.206	0.272	0.326
Se serum vs. head circumference	x	x	0.510	0.089	**0.587**	**0.021**
Se serum vs. APGAR score at 1 min.	x	x	0.046	0.887	−0.096	0.731
Se serum vs. APGAR score at 5 min.	x	x	0.252	0.428	0.211	0.449
Se serum vs. gestational age	x	x	0.058	0.857	0.142	0.612
GPX vs. body length	x	x	0.068	0.833	0.004	0.987
GPX vs. birth weight	x	x	−0.078	0.809	−0.053	0.848
GPX vs. chest circumference	x	x	0.056	0.861	0.051	0.855
GPX vs. head circumference	x	x	0.051	0.873	0.225	0.418
GPX vs. APGAR score at 1 min.	x	x	−0.194	0.544	−0.108	0.700
GPX vs. APGAR score at 5 min.	x	x	−0.092	0.776	−0.145	0.606
GPX vs. gestational age	x	x	0.066	0.837	0.395	0.144

T I—the first trimester of pregnancy; T II—the second trimester of pregnancy; T III—the third trimester of pregnancy; Se—selenium; GPX—glutathione peroxidase; *p* < 0.05. The bold numbers are statistically significant.

In the subgroup of pregnant women with pregnancy complications, a strong, statistically significant, negative correlation was found between Se intake in the second trimester and gestational age, and in the third trimester, a positive correlation between Se concentration in serum and head circumference.

In the remaining analyses, both for selenium intake, its concentration in serum and the GPX value, there was no correlation with the newborns’ birth measurements.

## 4. Discussion

Selenium is absorbed from food in the form of inorganic compounds like selenites and selenates or organic links—selenomethionine and selenocysteine. The absorption of selenium from organic compounds reaches 90–95%, whereas from inorganic links, it is lower by ca. 10% [6].

The recommended Se intake for the Polish pregnant population is 50 µg/d. Pregnant women in our study met these dietary recommendations in each trimester (Table 3). Comparable findings were reported by Holmquist et al. [8] (53 µg/day) for Norwegian pregnant women, whereas the intake in Chinese pregnant women was estimated at 30 µg/day [9]. It is important to note that selenium intake depends heavily on the selenium content of the soil, which affects its concentration in food. China, for instance, has particularly low soil selenium levels [10,11]. Dietary selenium intake significantly influences serum selenium concertation and GPX activity [12,13]. Both the concentration of Se in the serum and GPX activity are directly linked to the dietary intake of this element [12,13]. A marker of short-term changes in tissue selenium levels is the concertation in serum, while GPX activity in plasma and whole blood serves as a functional indicator of selenium status. According to the WHO, the optimal mean serum selenium concentration for healthy adults ranges from 39.5 to 194.5 µg/L, but to achieve maximum glutathione peroxidase activity, the optimal Se concentration is between 70 and 90 µg/L [14]. It is worth noting, however, that these values refer to the general adult population, not to pregnant women. There are no studies that would determine the optimal serum Se concentration in pregnant women to achieve maximum GPX activity.

As research results indicate, the concentration of selenium in the blood decreases significantly with subsequent trimesters of pregnancy [6,15]. In the participants of our study, a decrease in Se concentration was observed from 43.98 µg/L in the first trimester to 41.90 µg/L in the third trimester (Table 3). These concentrations are similar to those reported in other studies—40.5 μg/L (Hungary [16]); 38.21 μg/L (Turkey [17]); 51 μg/L (Yugoslavia [18]). The GPX values measured across the trimesters in our study were also consistent with those reported by other researchers [4].

Pregnant women with complicated pregnancies in our study had significantly lower serum Se concentrations compared to those with physiological pregnancy (Table 4). This increased selenium utilization may be a response to heightened redox processes associated with pregnancy complications, prompting an increase in GPX synthesis using the available selenium in the bloodstream (an insignificant increase in GPX concentration in this group compared to physiological pregnancies—Table 4).

The results for anthropometric measurements of newborns, as well as APGAR score at 1 and 5 min, showed no significant differences between groups with physiological and complicated pregnancy course (Table 4). These findings align with data from other European populations [19,20,21,22]. According to the WHO centile charts [23], the body lengths of all newborns we examined fell between the 75th and 90th percentile, body weights between the 50th and 75th percentiles, but head circumferences were only between the 10th and 25th percentiles.

Pregnancy duration differed significantly between the groups, with those experiencing complicated pregnancies having a higher average gestational age compared to those with physiological pregnancies (Table 4). It is worth noting that only one case of premature delivery was recorded in the group with physiological pregnancy, which influenced the average result in the group. Research by Barman et al. [24] on a large population of Norwegian pregnant women (72,025 women) showed that Se intake in the first half of pregnancy was significantly associated with a reduced risk of preterm birth and also influenced the duration of pregnancy.

The correlation analysis in our study revealed significant relationships between the first trimester selenium intake and both the weight and body length of newborns as well as Se-energy adjusted intake and the length, chest circumference of newborns and gestational age in the group of pregnant women with a physiological course of pregnancy (Table 5). Proper fetal development is influenced by genetic factors, fetal–placental circulation, environmental factors, and the nutritional status of the mother [25,26,27]. Among these, oxidative stress plays a crucial role in placental development and functional disorders [27,28]. Selenium is pivotal not only in maintaining oxidative balance but also in modulating immune responses, inflammatory processes, and cell apoptosis, which are crucial in the early stages of placental development [29]. Selenium also enhances duodenal function and affects the secretion of key hormones essential for early fetal development, such as thyroid hormones (TH), insulin-like growth factor (IGF) and insulin [30,31,32,33]. The authors of several studies have shown a positive correlation between the concentration of Se in the blood in early pregnancy and birth weight, or the risk of low birth weight [34,35,36].

In the second trimester, our study showed significant correlations between selenium intake, GPX activity and APGAR score at 1 and 5 min (Table 5). During this period of pregnancy, there is intensive development of the nervous system and myelination of nerve fibers, along with significant organ growth. Concurrently, there is an increase in lipid peroxidation products in maternal blood, which may decrease later in pregnancy [37]. Lipid peroxides, also produced in the placenta, can induce oxidative stress and significant cellar damage if uncontrolled [38]. In response, higher levels of antioxidant enzymes, including GPX, are activated. The undisturbed progression of developmental processes and antioxidant defense contributes to the proper functioning of essential systems such as the respiratory and circulatory systems immediately after birth, which are evaluated through APGAR scoring.

Also in the third trimester, a significant positive relationship was noted between GPX activity and newborn birth weight (Table 5). Other studies have similarly indicated that antioxidant status, including GPX activity, correlates with birth weight in newborns of appropriate gestational size [39,40].

The few correlations between selenium status and anthropometric parameters in our study may be due to the small size of the group. It can be assumed that with a large study population, there would be more similar correlations. Horan et al.’s study [41] showed a negative correlation between selenium intake in the first trimester and the ratio of the subscapular skin fold to the triceps fold in newborns. Similarly, in the third trimester, selenium intake was negatively correlated with the abdominal circumference of healthy pregnant newborns. Furthermore, in the study by Bizerea-Moga et al. [42], a positive correlation was demonstrated between the serum selenium concentration in normotensive and hypertensive pregnant women and the birth weight of the newborn. However, a study on the population of Polish newborns showed no significant correlations between the Se concentration in umbilical cord blood and any of the anthropometric parameters [43].

Fewer correlations between Se status and birth parameters were observed in the subgroup of women with complicated pregnancies. In this group, a significant negative correlation existed between Se intake and gestational age in the second trimester, and a positive correlation between serum Se concentration and head circumference in the third trimester (Table 5). Perhaps the less free Se there is in serum, the more of it is bound to GPX. Thanks to this, the pregnant woman’s body can maintain a redox balance for longer, extending the time until the physiological inflammation that initiates labor occurs. At the same time, the longer the pregnancy lasts physiologically, the greater the chances of the fetus achieving a normative head circumference.

Similar findings were reported by Lozano et al. [44] in a study of 1249 Spanish mother–newborn pairs, showing a negative correlation between serum Se concentration and pregnancy length, and a positive correlation with head circumference, but only among pregnant women whose serum selenium levels exceeded 15 µg/L. Unfortunately, the study did not specify whether any pregnancy complications occurred. The impact of selenium intake on pregnancy duration in the case of complications has not yet been extensively researched, and most studies focus on selenium concentration in serum or whole blood. However, a meta-analysis of these studies suggests a positive relationship between Se concentration and pregnancy duration [24,45].

In the case of pregnancy with complications, it is likely that the typical influences of selenium on fetal growth and development metrics may be overshadowed by the complexities introduced by these complications, indicating a potentially different metabolic or nutritional dynamic at play.

## 5. Strengths and Limitations

The study conducted has both its strengths and limitations. Its strength is the study design. The results were based solely on data obtained from pregnant women participating in the study from the first trimester of pregnancy to delivery. During this period, food samples corresponding to food diary entries were collected three times (once in each trimester). Blood and urine samples were also collected concurrently. This unique approach allowed for the assessment of actual selenium intake, unlike studies based only on estimated data from the FFQ.

The limitation of the study, however, is the low number of study participants, which could have translated into the results of statistical analyses and the high homogeneity of the studied group (education, place of residence and age). It can be assumed that with a large study population, there would be more correlations between selenium status and anthropometric parameters.

## 6. Conclusions

The results of our study highlight a discernible relationship between maternal selenium status during pregnancy—encompassing dietary intake, serum concentrations, and GPX activity—and various newborn anthropometric parameters, such as birth weight, body length, and APGAR scores. This correlation is particularly evident among women experiencing a physiological course of pregnancy, suggesting that optimal selenium levels play a significant role in promoting healthy fetal development in uncomplicated pregnancies.

However, the data also suggest that these correlations diminish or lose their predictive value in cases where pregnancy complications arise. In such scenarios, the typical influences of selenium on fetal growth and development metrics may be overshadowed by the complexities introduced by these complications, indicating a potentially different metabolic or nutritional dynamic at play.

This nuanced understanding underscores the need for targeted nutritional interventions and monitoring in pregnancies identified as high-risk or complicated. Further studies with larger, more diverse cohorts are needed to explore the differential impacts of selenium in various pregnancy scenarios to better tailor guidelines and recommendations for selenium intake during pregnancy. This could ultimately lead to more effective strategies for improving pregnancy outcomes and neonatal health through nutritional support, particularly in terms of selenium supplementation where it is most beneficial.

## Figures and Tables

**Figure 1 nutrients-16-02873-f001:**
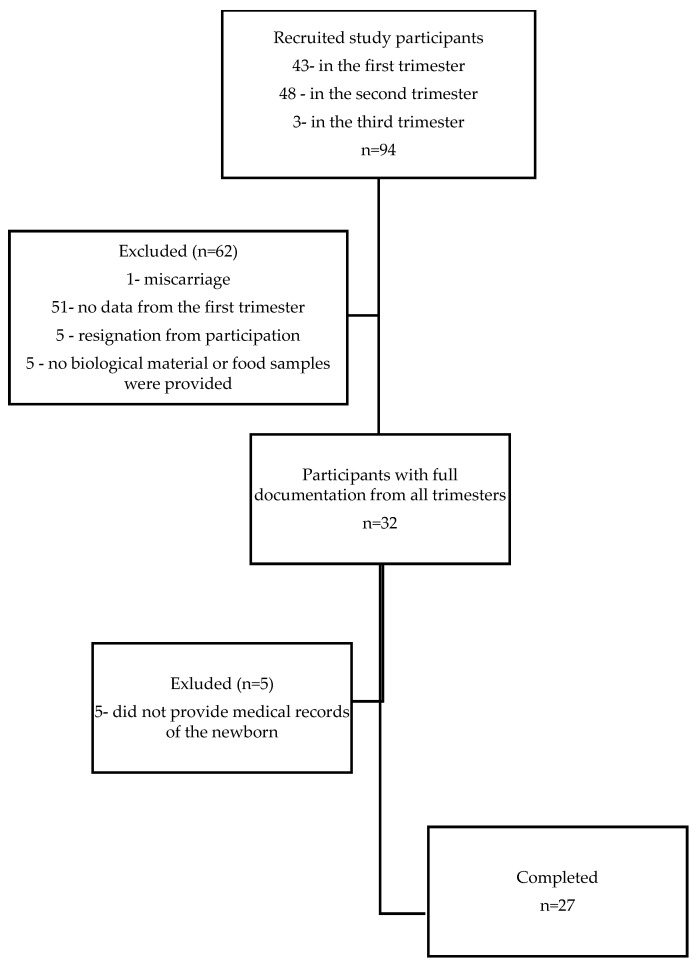
Participant flow chart.

**Table 1 nutrients-16-02873-t001:** Baseline characteristics of study participants in particular trimesters.

Demographics	T I	T II	T III
Pregnancy complications *n* (% of group)			
No	23 (85.2%)	15 (55.6%)	12 (44.4%)
Yes	4 (14.8%)	12 (44.4%)	15 (55.6%)
Hypothyroidism *	2 (7.4%)	4 (14.8%)	5 (18.5%)
Gestational diabetes mellitus **	1 (3.7%)	1 (3.7%)	3 (11.1%)
Urinary tract infections ***	1 (3.7%)	5 (18.5%)	3 (11.1%)
Anemia ****	-	2 (7.4%)	4 (14.8%)
Pregnancy (weeks) Median (Q1–Q3)	11 (9–12)	21 (17–23)	34 (29–36)
Weight gain (kg) Median (Q1–Q3)	2.64 (1.87–2.73)	6.18 (4.48–7.67)	5.58 (4.12–6.83)
Physical activity			
Sedentary	23 (85.2%)	19 (70.4)	26 (69.3%)
Moderately active	4 (14.8%)	8 (29.6%)	1 (3.7%)
Vigorously active	0 (0%)	0 (0%)	0 (0%)

* Subjects received medication (levothyroxine). ** None of the patients received insulin, and each of them received diet therapy. *** Subjects received medication (nitrofurantoin/furazidin). **** Subjects received iron supplementation.

**Table 2 nutrients-16-02873-t002:** Baseline characteristics of study participants in physiological and complicated pregnancy.

	Physiological Pregnancy *n* = 12	Pregnancy Complications *n* = 15	*p*-Value
Mean age (year)	30.0 ± 4.78	29.93 ± 4.95	0.314
Delivery *n* (% of group)			
Term delivery	12 (100%)	14 (93.3%)	1.000
Preterm delivery	0 (0%)	1 (6.7%)	
Mode of delivery *n* (% of group)			
Cesarean delivery	5 (42%)	2 (13%)	0.219
Vaginal delivery	7 (58%)	13 (87%)	
Birth weight			
<2500 g	2 (16.7%)	0 (0%)	0.188
>2500 g	10 (83.3%)	15 (100%)	
Education *n* (% of group)			
Elementary school	1 (8.3%)	0 (0%)	0.700
High school	0 (0%)	1 (6.7%)	
Academic	11 (91.7%)	14 (93.3%)	
Place of residence *n* (% of group)			
Urban	12 (100%)	14 (93.3%)	1.000
Rural	0 (0%)	1 (6.7%)	
Smoking cigarettes *n* (% of group)			1.000
Current smoker	0 (0%)	0 (0%)	
Quit smoking	3 (25%)	3 (20%)	
Never smoked	9 (75%)	12 (80%)	
Pre-pregnancy BMI *n* (% of group)			
BMI < 18.5	1(8.3%)	2 (13.3%)	0.034
BMI 18.6–24.9	11(91.7%)	8 (53.4%)	
BMI > 25	0 (0%)	5 (33.3%)	
Prenatal vitamin/mineral intake *n* (% of group)			
Vitamin/mineral supplements without Se	10 (83.3%)	9 (60%)	0.236
Vitamin/mineral supplements with Se	2 (16.7%)	6 (40%)	

## Data Availability

The original contributions presented in the study are included in the article, further inquiries can be directed to the corresponding author.

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
