# Peer review of "Is Maternal Selenium Status Associated with Pregnancy Outcomes in Physiological and Complicated Pregnancy?"

_nutrients, 2024, doi:10.3390/nu16172873_

Round 1

Reviewer 1 Report

Comments and Suggestions for Authors

In the manuscript Is Maternal Selenium Status Associated with Pregnancy Outcomes in Physiological and Complicated Pregnancy? the authors discussed the authors discuss maternal selenium status during pregnancy, including dietary intake, serum concentrations and glutathione peroxidase activity in relation to neonatal anthropometric parameters such as birth weight, length and APGAR score.

However, in my point of view, the manuscript should be improved.

1. What are the effects of vitamin/mineral supplements with or without Se on outcomes in complicated pregnancy?

2. Lines 157:  Different complications have different impacts on pregnancy outcomes, and simple comorbidities alone are not justified.

3. Lines 265-266: The decrease in serum selenium concentration from first trimester to third trimester but the increase in GPX activity require a more specific explanation.

4. Line 288: “ This difference may be linked to higher selenium intake in the group with complicated pregnancies (64 vs. 53 µg/day) , I don't think the description is quite right here because those experiencing complicated pregnancies have lower serum selenium concentrations.

5. The font size of table 2 needs to be standardised.

6. µg/l and U/l should be written as µg/L and U/L.

Comments on the Quality of English Language

none

Author Response

The comments and advices of Referees have helped us with correcting the mistakes and imprecision in the paper. I am very grateful for that.

Please find the new and altered passages highlighted in the manuscript in red letters.

The detailed description of the changes in the manuscript is listed below.

Answers to the Reviewer # 1:

  1. What are the effects of vitamin/mineral supplements with or without Se on outcomes in complicated pregnancy?

The aim of the study was to assess the effect of selenium status on neonatal outcomes, therefore, with such a small number of participants in the groups, the authors did not undertake additional analysis of the effect of supplementation. Nevertheless, it is a very interesting aspect, which we will include in our next project. Thank you for the suggestion.

  1. Lines 157: Different complications have different impacts on pregnancy outcomes, and simple comorbidities alone are not justified.

The authors are aware of the varied impact of individual complications on the course of pregnancy. However, due to the individual cases in each complication, it would not be possible to conduct statistical analyses, therefore the authors decided to combine all complications into one group.

  1. Lines 265-266: The decrease in serum selenium concentration from first trimester to third trimester but the increase in GPX activity require a more specific explanation.

The Discussion was supplemented with the following sentence:

„This increased selenium utilization may be a response to heightened redox processes associated with pregnancy complications, prompting an increase in GPX synthesis using the available selenium in the bloodstream (a insignificant increase in GPX concentration in this group compared to physiological pregnancies - Table 4).”

  1. Line 288: “ This difference may be linked to higher selenium intake in the group with complicated pregnancies (64 vs. 53 µg/day) ”, I don't think the description is quite right here because those experiencing complicated pregnancies have lower serum selenium concentrations.

After introducing the median instead of the mean in the data, the daily selenium intake in both groups turned out to be almost the same, so this sentence was removed from the Discussion.

  1. The font size of table 2 needs to be standardised.

The appearance of all tables has been unified.

  1. µg/l and U/l should be written as µg/L and U/L.

The indicated changes have been made.

Reviewer 2 Report

Comments and Suggestions for Authors

This is an interesting paper, as it tries to put light to a general aspect of nutrition with the essential trace element selenium. In spite of this, there are several aspects that should be improved or mentioned

Major

L. 122ff: While it is of major importance that the authors indeed quantified selenium intake of the included patients, rather than only deriving it from untrusty telefon questinairies, it should be highlighted what they determined by their technique, i. e. total selenium by atomic spectrometry rather than differentiating intake in the different forms as selenate, Se-methionine and Se-cysteine. This also applies to supplements (l.137).

L.77ff: The participating clinics should be indicated, and the whole available collective of eligible pregnant women in the participating clinics included in the flow diagram (Fig. 1).

How were the patients selected? Was it prospective and randomized? Specific data on medical care of patients like medication (e.g. insulin in gestational DM?) and the mode of delivery should be included.

There are many redundancies in the Results section. Data and p-values shown in tables mustn't be additionally provided in the text. Explain and refer to the respective table.

L. 154ff/Statistics: There is no mentioning of a power analysis to define required sample size of groups. Are the data corrected for multiple group comparisons? Otherwise it must be stated that this is an explorative pilot study and hypothesis generating. Moreover, for most data presented, medians, IQRs and full ranges will be much more helpful.

The high number of drop-out during the course of study have an impact on the value of data. It cannot be excluded that the drop-outs in quartiles 2 and 3 were all healthy so that the increase in complication rate is artificial. This must be discussed.

Tables should be concise, identical format, with font sizes not larger than used in the text. Similarly, in the figure, the letter size shouldn't be smaller, as it cannot be decipered otherwise.

Table 1 should be separated into 2, the second beginning from the line 'Physiological Pregnancy. Here, p-values should be included.

L. 191-193: The essence of data is that there are 2-6fold ranges of intakes and levels, and there is essentially no difference between groups. Replace this misleading wording and provide overall median/IQR/full range values in the text, in addition to the values of table.

Table 3: Is there a correlation between Se intake and its serum concentration? Explain, why Se intake on a total vs 1000kcal basis is antiparallel. Apparently, nutrition in the complications group was highly different, with more fat and sugar and a lower nutrient index. This may be a confounder to be discussed in the limitations of study.

Discussion:

The discussion is too long, and many paras can be shortened.Some parts are merely narrative rather than relating general and study data to the own findings.

Briefly discuss, how and in which forms Se is absorbed, and provide data on differential metabolism.

Rather than only mentioning 70-90µg/mL as target serum concentrations (l. 259), the authors should relate their and other data to these values

L. 267: >Despite a slightly higher dietary Se intake,...< This sentence is misleading. There was a high range of intake in either group, and no significant difference.

L. 303-329: This is all true, but much too wordy. Where is the relation to the study to be discussed here?

L. 351ff: This is in line with your findings, but what does it mean? Does it mean that brain growth is accelerated, and therefore a longer pregnancy isn't required? Discuss this!

Include a list of abbreviations

Minor:

Inclusion/exclusion parameters should be shown in a table.

L. 174: specify premature (weeks postconceptual age) and low birth weight (grams and percentile).

L. 177: specify 'almost 30%'.

L. 178ff: Provide meadin, IQR and full range of BMI, and specify the outlyers.

L. 203f: Refer to table, where the data are provided, rather than recapitulating, what's written there.

L. 216-227: Leave away all data and significance valiúes, as they are found in the table. Briefly (!) describe the results.

L. 245: These abbreviations should be introduced in a previous section. rather than in the discussion.

L. 288ff: What are the individual data of the preterm delivery in terms of Se-parameters etc.? The small number of study participants doesn't allow major conclusions here!

L. 293 and elsewhere: remove >statistically< in combination with significant. It's unnecessary. Every reader will understand that the word significant is meant in its statistics rather than relevance meaning.

L. 296: replace >...0.015) and chest...< by >...0.015), chest...<

L. 365: the first sentence is superfluous.

Comments on the Quality of English Language

The English is mostly fine. Some wording can be shortened.

Author Response

Major

  1. 122ff: While it is of major importance that the authors indeed quantified selenium intake of the included patients, rather than only deriving it from untrusty telefon questinairies, it should be highlighted what they determined by their technique, i. e. total selenium by atomic spectrometry rather than differentiating intake in the different forms as selenate, Se-methionine and Se-cysteine. This also applies to supplements (l.137).

We have updated the Selenium intake section with the following information

“The estimated total selenium intake (as an element in inorganic form) for each study participant was presented as an average over seven days.”

L.77ff: The participating clinics should be indicated, and the whole available collective of eligible pregnant women in the participating clinics included in the flow diagram (Fig. 1). How were the patients selected? Was it prospective and randomized? Specific data on medical care of patients like medication (e.g. insulin in gestational DM?) and the mode of delivery should be included.

There are many redundancies in the Results section. Data and p-values shown in tables mustn't be additionally provided in the text. Explain and refer to the respective table.

The study involved private obstetric clinics, which in Poland are usually sole proprietorships, where the clinic name includes the doctor's name. Unfortunately, I did not obtain consent from the cooperating doctors to publish their names. In addition, the doctors obtained information from the study's managers about the inclusion and exclusion criteria for the study and at this stage they made a selection among pregnant women and then forwarded them to the study authors. The study authors are unable to determine how many pregnant women did not meet the criteria or refused to participate in the study. Thanks to such selected participants, we were able to conduct a pilot prospective study, which is shown in the revised text of the manuscript. We have supplemented Table 1 with information on the treatment of pregnant women with complications. In the Results section we have removed from the text data and p-values.

  1. 154ff/Statistics: There is no mentioning of a power analysis to define required sample size of groups. Are the data corrected for multiple group comparisons? Otherwise it must be stated that this is an explorative pilot study and hypothesis generating. Moreover, for most data presented, medians, IQRs and full ranges will be much more helpful.

Tables should be concise, identical format, with font sizes not larger than used in the text. Similarly, in the figure, the letter size shouldn't be smaller, as it cannot be decipered otherwise.

Table 1 should be separated into 2, the second beginning from the line 'Physiological Pregnancy. Here, p-values should be included.

We have added a new section “Study Design” explaining how the sample size was calculated.

 “The minimum sample size was determined based on an α error probability of 0.5, a power of 0.8, and a minimum effect size of 0.5 for correlations test, yielding a required sample size of 21 patients. Eventually, 27 pregnant women could be recruited for the study based on the inclusion and exclusion criteria (see Figure 1). Post hoc power analysis for the correlation test, assuming a minimum effect size of r = 0.5, an α error probability of 0.05, and a sample size equal to 12 for the physiologic pregnancy group, yielded a power of 0.68. For the complicated pregnancy group of 15 with the same assumptions, the power of analysis was 0.59. These calculations were performed using the online G*Power calculator (v. 3.1.9.6, developed by Franz Faul, Universitas Kiel, Germany, https://www.psychologie.hhu.de/arbeitsgruppen/allgemeine-psychologie-und-arbeitspsychologie/gpower).”

In Tables 1,3,4 the results are presented as median (Q1-Q3).

Table format and letter size have been unified.

Table 1 has been divided into two separate parts and the p-value is provided in the new Table 2.

  1. 191-193: The essence of data is that there are 2-6fold ranges of intakes and levels, and there is essentially no difference between groups. Replace this misleading wording and provide overall median/IQR/full range values in the text, in addition to the values of table.

Table 3: Is there a correlation between Se intake and its serum concentration? Explain, why Se intake on a total vs 1000kcal basis is antiparallel. Apparently, nutrition in the complications group was highly different, with more fat and sugar and a lower nutrient index. This may be a confounder to be discussed in the limitations of study.

The introduction of the median instead of the average values ​​resulted in the results of selenium supply (per day and in terms of energy) being comparable in both study groups. For this reason, the introduction of information about a different way of nutrition in the group with pregnancy complications could be misleading for the reader.

In our previous study, we showed that selenium intake correlates positively with its serum concentration. (Pieczyńska, J.; Płaczkowska, S.; Sozański, R.; Orywal, K.; Mroczko, B.; Grajeta, H. Is maternal dietary selenium intake related to antioxidant status and the occurrence of pregnancy complications?. J. Trace Elemen. Med. Biol. 2019, 54, 110-117.)

The discussion is too long, and many paras can be shortened. Some parts are merely narrative rather than relating general and study data to the own findings.

Briefly discuss, how and in which forms Se is absorbed, and provide data on differential metabolism.

Rather than only mentioning 70-90µg/mL as target serum concentrations (l. 259), the authors should relate their and other data to these values.

The authors have shortened the Discussion section to include general information.

 Regarding Se forms and metabolism, the following text has been added:

“Selenium is absorbed from food in the form of inorganic compounds like selenites and selenates or organic links– selenomethionine and selenocysteine. The absorption of selenium from organic compounds reaches 90–95%, whereas from inorganic links, it is lower by ca. 10% [6].”

 Information on GPX activity in pregnancy has also been added:

“]. It is worth noting, however, that these values refer to the general adult population, not to pregnant women. There are no studies that would determine the optimal serum Se concentration in pregnant women to achieve maximum GPX activity.”

  1. 267: >Despite a slightly higher dietary Se intake,...< This sentence is misleading. There was a high range of intake in either group, and no significant difference.

The sentence has been corrected.

  1. 303-329: This is all true, but much too wordy. Where is the relation to the study to be discussed here?

The text has been shortened. The authors hope that now the connection between Se and GPX concentrations in early pregnancy and fetal development will be clearer.

  1. 351ff: This is in line with your findings, but what does it mean? Does it mean that brain growth is accelerated, and therefore a longer pregnancy isn't required? Discuss this!

We have updated the Discussion section with the following information:

“Perhaps the less free Se there is in serum, the more of it is bound to GPX. Thanks to this, the pregnant woman's body can maintain redox balance for longer, extending the time until the physiological inflammation that initiates labor occurs. At the same time, the longer the pregnancy lasts physiologically, the greater the chances of the fetus achieving a normative head circumference.”

Include a list of abbreviations

We have placed a list of abbreviations at the beginning of the manuscript.

Inclusion/exclusion parameters should be shown in a table.

  1. 174: specify premature (weeks postconceptual age) and low birth weight (grams and percentile).
  2. 177: specify 'almost 30%'.
  3. 178ff: Provide meadin, IQR and full range of BMI, and specify the outlyers.
  4. 203f: Refer to table, where the data are provided, rather than recapitulating, what's written there.
  5. 216-227: Leave away all data and significance valiúes, as they are found in the table. Briefly (!) describe the results.
  6. 245: These abbreviations should be introduced in a previous section. rather than in the discussion.
  7. 288ff: What are the individual data of the preterm delivery in terms of Se-parameters etc.? The small number of study participants doesn't allow major conclusions here!
  8. 293 and elsewhere: remove >statistically< in combination with significant. It's unnecessary. Every reader will understand that the word significant is meant in its statistics rather than relevance meaning.
  9. 296: replace >...0.015) and chest...< by >...0.015), chest...<
  10. 365: the first sentence is superfluous.

Since information regarding the inclusion and exclusion criteria was presented in the text and Figure 1, the authors decided not to duplicate the data in a separate table.

With respect to the remaining comments, the authors made the necessary corrections in the text and tables.

Reviewer 3 Report

Comments and Suggestions for Authors

The manuscript is generally well-organized, with a logical flow from the introduction through to the discussion. The language used is clear, concise, and appropriate for an academic audience, making the manuscript accessible to readers with varying levels of expertise. The abstract effectively summarizes the key findings of the study, providing a clear overview of the research objectives, methods, results, and conclusions. The study design is clearly defined with a focus on assessing selenium intake, serum concentration, and GPX activity across different trimesters of pregnancy. The study attempts to explore a novel correlation between maternal selenium status and neonatal outcomes, which is commendable. The results are presented in a clear and organized manner, with appropriate use of tables to summarize key findings.

However, there are certain areas that need to be addressed:

·      The sample size appears small (only 27 participants completing the study), which limits the statistical power of the findings. Consider discussing the potential impact of this limitation more thoroughly in the discussion section.

·      The inclusion criteria exclude women with pre-existing conditions, which is appropriate. However, the exclusion of multiple pregnancies may overlook important variables. It might be useful to justify why these were excluded and discuss how this might affect the generalizability of the results.

·      The use of a seven-day food diary for dietary selenium intake assessment is appropriate, but food diaries can be prone to recall bias. The authors should discuss potential biases in the methodology section and how they were addressed.

·      The authors do not explicitly mention the use of control groups. Including a control group with no selenium supplementation or a placebo group would strengthen the study's validity.

·      Page 7, table 3: The presentation of neonatal outcome measurements and selenium status is clear, but the text does not always clearly align with the tables. For example, the significance of differences between groups is mentioned, but the corresponding p-values are sometimes not emphasized in the text. Consider highlighting these more explicitly to guide the reader through the findings.

·      The interpretation of the results, particularly the correlation between selenium intake and neonatal outcomes, is somewhat limited. The manuscript should provide a deeper analysis of what these correlations mean in the context of existing literature. For instance, discussing whether these correlations are biologically plausible and how they compare with findings from other studies would add depth to the analysis.

·      The manuscript focuses on correlations that are statistically significant but does not adequately address the lack of significant findings in other areas. It's important to discuss these non-significant results to provide a balanced view of the study's findings.

·      Given the small sample size, the study likely lacks sufficient statistical power to detect smaller effect sizes. This is particularly problematic in the subgroup analyses, Page 8, Lines 214-233. Show the power analysis determining the necessary sample size in methodology.

·      There are instances of redundancy in the text. For example, certain points are repeated in both the results and discussion sections without adding new insights. Streamlining these sections to avoid repetition will enhance the clarity and impact of the manuscript.

·      Some sentences are overly long or complex, which can make them difficult to follow. For instance, the discussion on the strengths and limitations of the study could be more concise. Breaking down long sentences into shorter, more direct statements would improve readability.

·      “Selenium content” should be “selenium concentration”.

Overall, the study explores an important topic with a novel approach, but its impact is limited by a small sample size and potential methodological biases. The study's findings are interesting but should be interpreted with caution due to these limitations.

Author Response

The comments and advices of Referees have helped us with correcting the mistakes and imprecision in the paper. I am very grateful for that.

Please find the new and altered passages highlighted in the manuscript in red letters.

The detailed description of the changes in the manuscript is listed below.

Answers to the Reviewer # 3:

   The sample size appears small (only 27 participants completing the study), which limits the statistical power of the findings. Consider discussing the potential impact of this limitation more thoroughly in the discussion section.

Given the small sample size, the study likely lacks sufficient statistical power to detect smaller effect sizes. This is particularly problematic in the subgroup analyses, Page 8, Lines 214-233. Show the power analysis determining the necessary sample size in methodology.

We have added a new section “Study Design” explaining how the sample size was calculated.

 “The minimum sample size was determined based on an α error probability of 0.5, a power of 0.8, and a minimum effect size of 0.5 for correlations test, yielding a required sample size of 21 patients. Eventually, 27 pregnant women could be recruited for the study based on the inclusion and exclusion criteria (see Figure 1). Post hoc power analysis for the correlation test, assuming a minimum effect size of r = 0.5, an α error probability of 0.05, and a sample size equal to 12 for the physiologic pregnancy group, yielded a power of 0.68. For the complicated pregnancy group of 15 with the same assumptions, the power of analysis was 0.59. These calculations were performed using the online G*Power calculator (v. 3.1.9.6, developed by Franz Faul, Universitas Kiel, Germany, https://www.psychologie.hhu.de/arbeitsgruppen/allgemeine-psychologie-und-arbeitspsychologie/gpower).”

We have also added a new sentence in section “Study Limitation”

“It can be assumed that with a large studied population there would be more correlations between selenium status and anthropometric parameters.”

The inclusion criteria exclude women with pre-existing conditions, which is appropriate. However, the exclusion of multiple pregnancies may overlook important variables. It might be useful to justify why these were excluded and discuss how this might affect the generalizability of the results.

We have added a new sentence in section “Participants”:

“ Women with multiple pregnancies were also excluded from the study. Having more than one fetus creates a higher metabolic demand than a single pregnancy, which may increase the risk of nutrient deficiencies.”

The use of a seven-day food diary for dietary selenium intake assessment is appropriate, but food diaries can be prone to recall bias. The authors should discuss potential biases in the methodology section and how they were addressed.

The section “Selenium Intake” was supplemented with a sentence:

“To avoid memory errors, each record was verified by a trained interviewer in terms of composition and quantity of recorded products and cross-referenced with the food samples provided.”

The authors do not explicitly mention the use of control groups. Including a control group with no selenium supplementation or a placebo group would strengthen the study's validity.

Because the study was prospective in nature, a control group was not included. Nevertheless, we will use this suggestion in the design of the next study.

Page 7, table 3: The presentation of neonatal outcome measurements and selenium status is clear, but the text does not always clearly align with the tables. For example, the significance of differences between groups is mentioned, but the corresponding p-values are sometimes not emphasized in the text. Consider highlighting these more explicitly to guide the reader through the findings.

The description of the results has been reworded.

The interpretation of the results, particularly the correlation between selenium intake and neonatal outcomes, is somewhat limited. The manuscript should provide a deeper analysis of what these correlations mean in the context of existing literature. For instance, discussing whether these correlations are biologically plausible and how they compare with findings from other studies would add depth to the analysis.

The manuscript focuses on correlations that are statistically significant but does not adequately address the lack of significant findings in other areas. It's important to discuss these non-significant results to provide a balanced view of the study's findings

The discussion in this area has been improved. The following sentence has been added:

“The few correlations between selenium status and anthropometric parameters in our study may be due to the small size of the group. It can be assumed that with a large studied population there would be more such correlations. Horan et al.'s study [41] showed a negative correlation between selenium intake in the first trimester and the ratio of the subscapular skin fold to the triceps fold in newborns. Similarly, in the third trimester, selenium intake was negatively correlated with the abdominal circumference of healthy pregnant newborns. Also in the study by Bizerea-Moga et al. [42] a positive correlation was demonstrated between the serum selenium concentration in normotensive and hypertensive pregnant women and the birth weight of the newborn. However, a study on the population of Polish newborns showed no significant correlations between the Se concentration in umbilical cord blood and any of the anthropo-metric parameters [43].”

There are instances of redundancy in the text. For example, certain points are repeated in both the results and discussion sections without adding new insights. Streamlining these sections to avoid repetition will enhance the clarity and impact of the manuscript.

We hope that shortening the discussion has helped to avoid repetitions.

Some sentences are overly long or complex, which can make them difficult to follow. For instance, the discussion on the strengths and limitations of the study could be more concise. Breaking down long sentences into shorter, more direct statements would improve readability.

Long, complex sentences have been reworded into shorter form.

 “Selenium content” should be “selenium concentration”.

Changes have been made to the text.

Round 2

Reviewer 3 Report

Comments and Suggestions for Authors

I appreciate authors for revising the manuscript and considering all my suggestions.